# Parametric and Visual Programming BIM Applied to Museums, Linking Container and Content

**Massimiliano Lo Turco \*, Elisabetta Caterina Giovannini and Andrea Tomalini**

Department of Architecture and Design, Politecnico di Torino, 10125 Torino, Italy;
elisabettacaterina.giovannini@polito.it (E.C.G.); andrea.tomalini@polito.it (A.T.)
\* Correspondence: massimiliano.loturco@polito.it

**Abstract:** In recent years we have been experiencing an ever-increasing number of Building Modeling Modeling (BIM) and Visual Programming Language (VPL) approaches in the architectural design field. These experiments have inspired new research strictly focused on exploring values, criticalities, and the advantages of applying these combined methodologies in the Cultural Heritage domain. This integrated approach has emphasized the benefits derived from HBIM. The next step is to critically evaluate the application of BIM and VPL processes used in the management and valorisation of museum heritage, pursuing both parametric and algorithmic approaches. The research group worked on building a model that shared the BIM hierarchical structure and the flexibility of the VPL methodologies. Semi-automatic procedures were developed within a rigorous BIM workflow, with the help of Autodesk and McNeel tools, to show and manage complex museum management phenomena. These procedures aimed to respond to three different objectives. First, the need to associate information from the Facility Report to the individual BIM components to predict and monitor the conditions in which museum collections are found. Second, the intention to measure the attractiveness of the artifacts within the exhibition project and the design effects for a correct prefiguration of visitor flows. Third, the elements involved included the exhibition area obtained from an HBIM model (converted into a visual field through interoperable processes), the digitized collections (the attractive elements), the users and, finally, the numerical evaluation of the visibility of specific objects within collections by simulating the human point of view. Once automated, the devised procedures can be considered a prototype to support curators in controlling and improving the efficiency of the exhibition layout.

**Keywords:** HBIM; integrated digital model; Visual Programming Language; digital cultural heritage; museum

## 1. Introduction

In the last decades the use of digital technologies has completely changed and improved the working methods (from 3D acquisition to the representation and modeling phases) in the architectural heritage domain. With specific regard to the 3D modeling object-based approach, one of the most meaningful definitions of Building Information Modeling reported in international standards is a "shared digital representation of physical and functional characteristics of any built object [. . .] which forms a reliable basis for decisions" [1]. In the heritage (HBIM) field, the digital reconstruction of complex shapes seems to be a very challenging task. Once having obtained the point cloud and identified the single elements and their mutual relationships, the user can:

- build an in-place family directly in the project environment
- create building components that can be reused in other projects (usually BIM platforms do not allow point clouds to be imported into the Family Editor except when using specific plugins)

- create 3D objects in another software package and import them into the BIM model as surface models.

These three approaches have different levels of complexity and produce qualitatively different outcomes. In this regard, it is worth critically reflecting on the interpretation of a rigorous BIM approach, meaning the complete exploitation of BIM approaches for Cultural Heritage buildings not only in terms of geometric accuracy, but also with consideration for the quality and accuracy of information associated with the building components [2] (Figures 1 and 2).

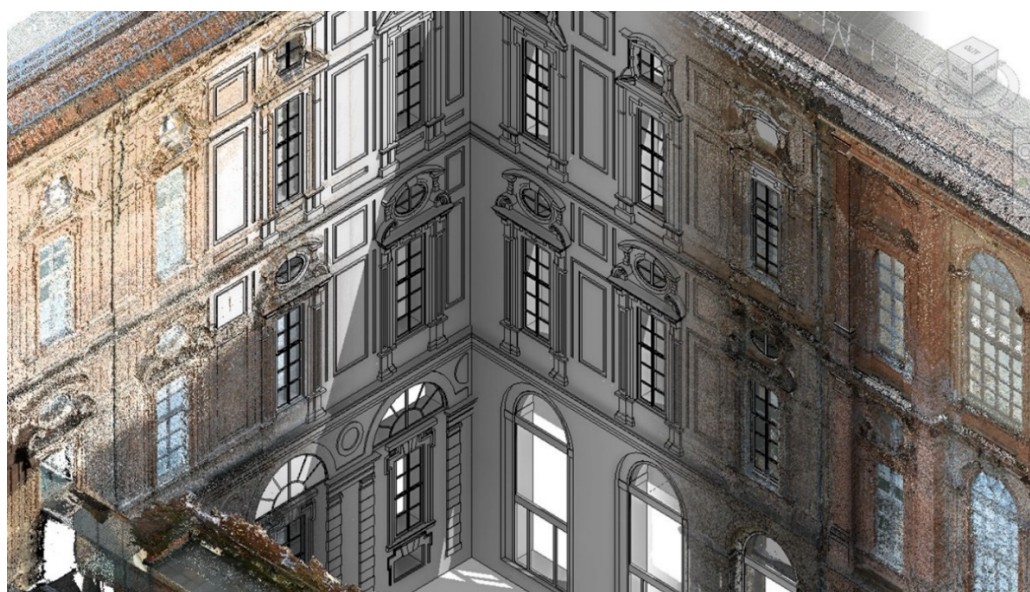

**Figure 1.** From the point cloud to HBIM model. Virtual model of the Fondazione Museo delle Antichità Egizie di Torino.

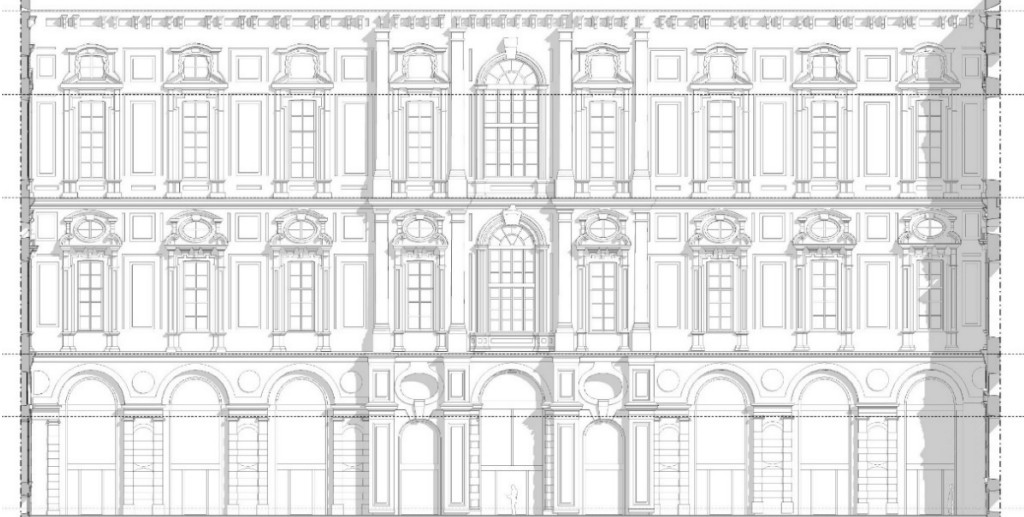

**Figure 2.** Internal facade of the Collegio dei Nobili. Image by the authors.

Before discussing the possible connections between museums and collections, it is convenient to summarize some taxonomies related to the digital modeling of complex artifacts belonging to the Cultural Heritage field. According to the HBIM type, it is essential to declare the purposes and the workflow adopted to make the process of infographic modeling implementable and reusable, defining the advantages and the limits in relation to

the efforts needed and the expected use of the model. In this regard, we describe available practical scenarios, not excluding mixed solutions that provide calibrated hybridization.

(a) Global vs. local model: the need to model the entire artifact versus other workflows that involve the digitization of ad hoc semantic-aware libraries.

(b) Parametric vs. non-parametric approach: the building components can be modeled using native components, identifying variants and invariants and progressively defining virtual objects through repeated nesting processes to obtain parametric objects, geometrically defined. This makes it possible to state that the model's elements are parametric objects, geometrically modifiable and reusable (Figure 3). A different procedure can be used to model non-conventional components, such as those not directly related to a specific model category (i.e., the vaulted systems). In these cases, interoperable processes between mathematical modelers and algorithmic platforms can be used for a very reliable geometric representation of the elements, sometimes at the expense of more rigid management of the informative apparatus associated with the native elements available in the BIM environment, as described in the following taxonomy.

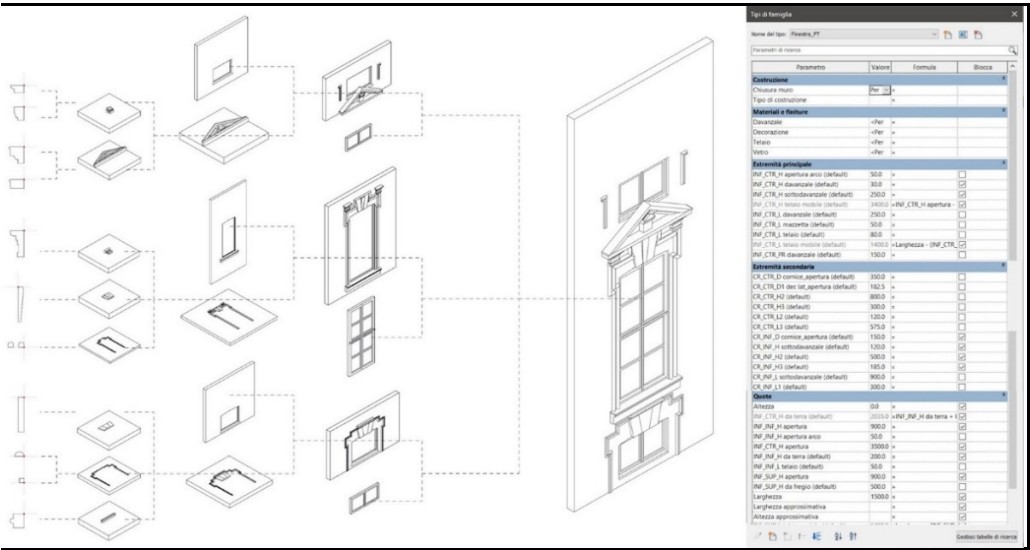

**Figure 3.** Nesting procedures for the building component collection: local models and geometric parameterization. Image by the authors.

(c) Geometric Modeling vs. Information Modeling: according to the aims of the digital modeling process, it is necessary to consider the definition of geometric contents (LoG—Level of Geometry) and information attributes (LoI—Level of Information), which are helpful for a more comprehensive description of the artifact for the subsequent stages of design and/or maintenance of the asset, as required by current legislation.

In this essay, we go beyond the goals of an HBIM approach exclusively oriented toward the informed digitization of the building-container-museum site to relate the information apparatus contained therein with the data associated with the museum collections (contents) when properly digitized and informed. The possible correlation between these two macro elements, carried out through interoperable approaches that also include the adoption of Visual Programming systems, can contribute to the effective management of a digital twin that can be used for different purposes. Dissemination and sharing with the public, evaluation of possible temporary museum installations, and the possibility of prefiguring possible flows of people, in an attempt to optimize the paths in complete safety, are the main aspects analyzed by these innovative workflows born from the interpolation of these methodologies.

## 2. HBIM Approaches for Museums: The State of the Art

Furthermore, in the museum field the adoption of BIM methodologies can take on different connotations, as classified below:

(a)   Building Information Modeling for newly built museums: the first taxonomy includes new design interventions characterized by a high level of complexity, but these procedures do not diverge from those adopted for different uses. Among the most exemplary cases, we can mention the Grand Egyptian Museum in Giza, designed by Heneghan Peng Architects (BIM model realized by DVA) (Figure 4), and the new Museum of the Future in Dubai by Killa Design, a project with a particularly futuristic and toroidal shape, fully managed in three dimensions, from conceptualization to construction management.

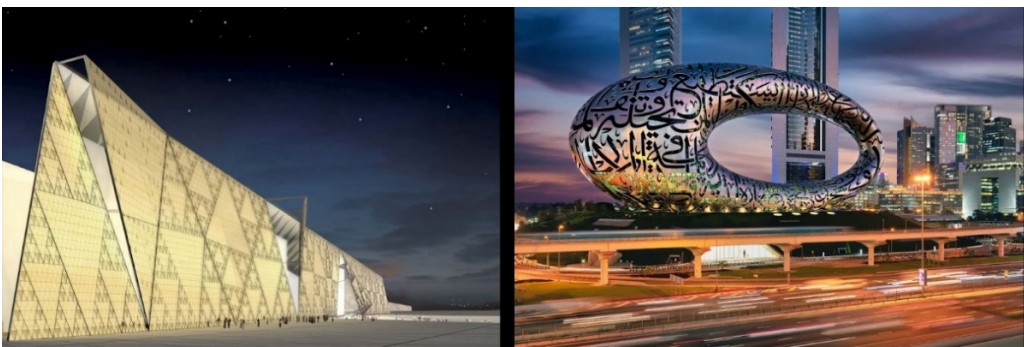

**Figure 4.** Samples of major museums designed with parametric approaches. On the (**left**), the Grand Egyptian Museum in Giza, designed by Heneghan Peng Architects (BIM model realized by DVA). On the (**right**), the new Museum of the Future in Dubai by Killa Design.

(b)   HBIM models for museums located in archeological sites/historic buildings: the second taxonomy includes all the typical criticalities of an HBIM approach, from the complexity of the survey operations to the modeling of elements characterized by a high degree of complexity and a low level of standardization [3,4].

One case study to be examined is the computerized digitization of the Borghese Gallery, carried out in collaboration with the Department of Architecture of the University of Rome La Sapienza. The laser-scanner point clouds of all the acquired marble surfaces will form the basis for the construction of a BIM model for Facility Management purposes. This model will monitor the MEP components and analyze the safety level of the internal environment and, consequently, the environmental conditions to which the exhibited works are subjected [5].

The examples of museums and buildings of architectural heritage managed through the BIM approach are numerous; however, fewer app licenses are related to the archaeological field. Among these, the work carried out in the Archaeological Park of Pompei stands out, with the first survey activities using laser scanners in the Domus of Arianna and the subsequent conversion of data into HBIM servers, creating a database to be interrogated to strategically plan maintenance operations. Such a database can provide concrete monitoring of the decays of the building, including structural ones, through a computerized comparison between the data entered in the digital model and the same data detected in real time by the sensors. The results have led to more orderly and structured intervention planning. The project is carried out by the Pompeii Archaeological Park together with the Federico II Università di Napoli, the Politecnico di Milano, and the Institute of Cultural Heritage Sciences of the CNR (National Research Council).

(c)   Information Modeling applied to buildings and collections: the latter taxonomy refers to the more mature experiences of complete digitalization of the physical museum implemented through effective integration with the digitized artifacts of the museum.

An interesting example concerns the implementation of the BIM model of the Galleria dell'Accademia, in Florence [6]. It is a museum information system based on BIM (hence Museum BIM-MBIM) connected to external databases. The research also involves digitizing a part of the collections for transversal management of the information relating to the building and the works from a graphic and informative point of view. This approach simplifies the management of the procedures prescribed by international best practices (such as Facility Report and Condition Report for the loan of works of art) and the verification of the compliance of a museum with minimum standards.

A second exciting case study is the development of the temporary exhibitions at the Nationalmuseum in Stockholm. A BIM model was used to support curators and conservators in critically evaluating different museum layout scenarios. Further research in this direction has been conducted within the European SensMat project. The first results have seen the construction of museum BIM models and the implementation of IoT sensors for monitoring collections [7].

The research that comes closest to that undertaken by the authors refers to the activity carried out by the researchers of the Department of Civil Engineering and Architecture (DICar) of the University of Catania. This research proposes a new operational methodology, called HS-BIM (Historical Sentient-Building Information Model) [8], which aims to implement HBIM models through the use of Artificial Intelligence (AI) techniques, applying the medical approach used in the disciplines of restoration and conservation (the factory as a living organism) as part of the HBIM methodology. This proposal was applied to the villa Zingali Tetto (Catania, 1930), home to the Museum of the Represiamento (MuRa). The study aims to conserve museum collections, particularly thermo-hygrometric conditions, identifying similarities between the physical and behavioral components highlighted in living beings and the tools used in BIM-based methodologies. For example, the ability to receive real-time data of different natures from diagnostic sensors placed in strategic points of the building as if it were a peripheral nervous system receiving external stimuli. In this configuration, despite the complexity of the processes put in place, there is still no dynamic behavior on the part of the model. By creating a decision support system (DSS) based on Artificial Intelligence mechanisms, the BIM model of the building can assume a synthetic attitude in the processing of stimuli, thus becoming "sentient", hence the transition from H-BIM to HS-BIM. Thanks to VPL computational techniques, the prototype can analyze, compare, manage, catalogue, and reorder the data and the various relationships contained in the models [9]. Other similar approaches are carried out with the aid of traditional programming systems that communicate not with non-BIM information models but with a cloud-based platform. For example, Alsuhly and Khattab developed an IoT-based system for monitoring and controlling a museum's internal environment. In this example, information about the environment, artifacts, and their security conditions are collected in real time and sent to a gateway where they are pre-processed and aggregated before being forwarded to a cloud where they are stored and analyzed. This approach allows automatic action by the actuators to minimize risk situations. Manual actuators are, however, provided for museum managers [10].

## 3. The Proposed Solutions

As anticipated above, recent years have seen the growing adoption of the integrated use of BIM and VPL methodologies for new construction or interventions on contemporary buildings. These interoperable processes have inspired new research describing the impacts on historic built heritage.

The research team focused on the modeling of the information attributes related to both the building and the elements it contains. For structuring the frameworks explained below, we opted to use Revit and Rhino3D software and their Visual Programming Languages: Dynamo and Grasshopper. They were chosen because they are a relatively widespread reference standard for the construction sector. Moreover, their interoperable nature makes them suitable for storing different information and communicating with other applications

or resources. The close relationship between container and content in museums, pursued through original and little investigated interoperable processes, can be considered an innovative element [11]. This relationship can assume different meanings due to increasing levels of complexity.

In particular, this relationship was investigated during the development of three different workflows. The theoretical and methodological considerations find their application in the digitization of spaces and collections related to the temporary exhibition "Invisible Archaeology" of the Egyptian Museum of Turin, which is fertile ground for many experiments carried out in collaboration with the Department of Architecture and Design of the Politecnico di Torino and the museum itself.

1.  The first research activity dealt with the problems linked to the growing need to create infographic protocols for the documentation and management of architectural heritage, organizing the information apparatus related to Cultural Heritage in a virtuous way. It is possible to define interconnected tools for the management and enhancement of heritage by providing multidisciplinary and interoperable information conversion processes and acquiring metric, geometric and material data for museums and their collections. The relationship between container-museum (Building Sheet/Environment Sheet) and content-collection (Object Sheet) is qualified in the connection between the information systems produced [12], with obvious management benefits [13]. The starting point for preparing these workflows was to systematize container and content through the Facility Report. The Facility Report is a technical document that describes the characteristics and plant equipment of the exhibition halls and warehouses to ensure adequate conservation of the works and prepare the Safety and Emergency Plan (PSEM) of the Museum. The data necessary to compile the Facility Report are introduced by the Environmental Charts, which is provided by the UNI 10829:1999 standard entitled "Goods of historic, artistic interest-Environmental conservation conditions-Measurement and analysis" [14]. Environmental maps are required to record the microclimate and lighting conditions of the spaces used to conserve museum collections. To date, these documents take the form of technical reports. However, they do not promote effective and efficient procedures to correlate geometric and morphological attributes of the architectural structure with descriptive performance, estimative data, etc. The Facility Report is usually employed in the design phase, more rarely for scheduled maintenance or preservation operations (Figure 5).

2.  The second research effort investigated the possibility of prefiguring the flow of visitors within museum environments. It is important to remember how the collections displayed in the setup spaces—the content of the museum—are recognizable entities thanks to formal and aesthetic attributes that recall, in the mind of the observer, invisible properties. Those attributes, together with the previous ones, give an attractive weight to the artworks. By attractive weight, we mean, for example, the historical, patrimonial, and media value but also the attractive value derived from previous surveys able to identify the interest that a single work had on visitors during previous installations. Furthermore, to prefigure the natural flow of visitors in the museum environment, it was necessary to integrate the simulation model of the rules that reproduce the behavioral characteristics created between the user, the works, and the environment set up in which the visitor is inserted. The behavioral profile used in this work derived from the classification of visiting styles developed in the ethnographic field by the scholars Eliseo Veron and Martine Levasseur [15]. In this context, the definition of criteria for the design and management of installations and user flows can take place through integrated information systems. The development of dynamic models can be a helpful tool to evaluate alternative scenarios to optimize flows for safety reasons [16]. BIM platforms are specifically designed for the semantic enrichment of digital building models, but these procedures need to be partially rethought for geometric/information modeling of collections [17]. Visual

Programming Language (VPL) methodologies, due to their flexible and interoperable nature with BIM systems, have been employed to create enriched models that can simulate human behavior while taking into account usually unrepresented data. To do so, it is necessary to introduce force fields, entities that usually belong to the world of physics. A force field describes the presence of a force applied to each point in a geometric domain. It is a function that connects each position to a vector with the force's intensity and direction. The charge intensity is a defined parameter strictly related to the characteristics of the associated metadata. Decay represents the rate at which the charge effects generated by a point entity decrease within the field. Thus, points with high charge values generate a larger attractive force field (Figure 6).

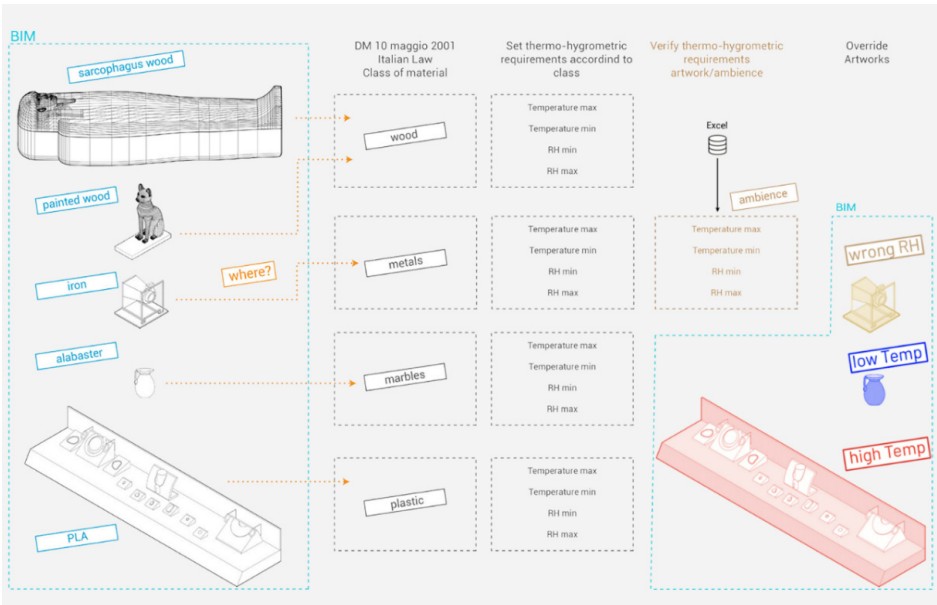

**Figure 5.** Scheme for verifying the environmental suitability between the artworks and the exhibition environment. Image by Arch. M. Cammarano.

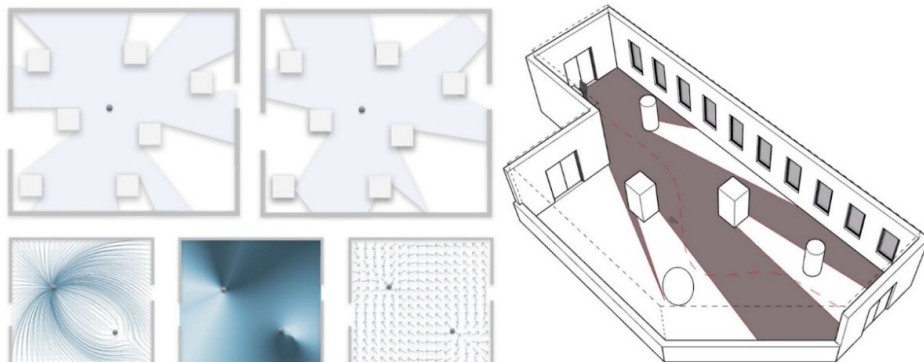

**Figure 6.** Top left: graphic representation of the isovist, a variable used to calculate the amount of space visible from any point within an environment. Below: diagrams representing the intensity of the charge generated by a point entity, in relation to the characteristics of the associated metadata. Right: spatial visualization of the isovist and the path of a visitor based on the attractive power associated with the collections.

3. The third research activity engaged the research team in identifying a tool that could simulate the visual experience and calculate the visibility of certain elements within the exhibit. The input properties for this type of tool are resolution and viewing radius. The resolution is similar to the characteristic of human vision called visual acuity,

which is the ability of the human eye to receive as much detail as possible. The vision radius is close to the sharpness of vision, hence the ability to perceive distant objects. This parameter is governed by the size of the visual radius length, which allowed us to define the amount of space perceived by the user. The visitor behavior described in this paper derived from the adoption of the classification of visiting styles elaborated in ethnography studies by scholars who grouped the different types of visitors into three categories: (i) taking into account the time devoted to the visit, (ii) the movement in museum exhibitions, and (iii) the attention to the single artwork belonging to the museum collection [15]. The case study links the graphic and informative apparatus of environments and collections to the temporary exhibition "Invisible Archaeology", located on the third floor of the Accademia delle Scienze building. The main exhibition room, the exhibition mezzanine, and the oval access to the exhibition were created after the latest restoration carried out by the Isolarchitetti studio to refurbish the rooms of the Pinacoteca Reale. The exhibition is particularly interesting because it combines real objects and digital artifacts, illustrating the documentation processes of the excavations. It continues with a section on diagnostic analyses and, finally, it focuses on "showing the invisible": from the jewels preserved inside the mummies (virtually reconstructed and 3D printed) and the analysis of the composition of the pigments used to decorate the sarcophaguses to the effective use of video mapping for an extraordinarily original and involving multimedia narrative experience. In particular, it must be stressed that it is not easy to combine the physical-technical maintenance and exhibition requirements of the collections under the constraints of historical architecture. In the case of temporary exhibitions, it is necessary to consider further requirements since the binomial container-content relationship constitutes a strongly interconnected system. The choice of specific artworks strongly influences the analysis, the management, and the methods of preservation of the objects themselves; moreover, the setting up of exhibition spaces aimed at reaching qualitative levels is in line with the standards of museum environments concerning microclimatic conditions, particularly with what is reported in section VI of the Ministerial Decree 10 May 2001 regarding the Management of Collections [18]. This is the context that composes the reference framework in which the research group worked. Taking into consideration the issues of conservation and protection of heritage, we have taken the opportunity to derive and manage fundamental information for the reconstruction of collections and exhibition elements in a digital environment, using the full capabilities of the BIM methodology. The procedure involved a tool that can support registrars and curators, who are responsible for the layout definition task used to optimize and facilitate the design phase and the management of artworks in an exhibition (Figure 7).

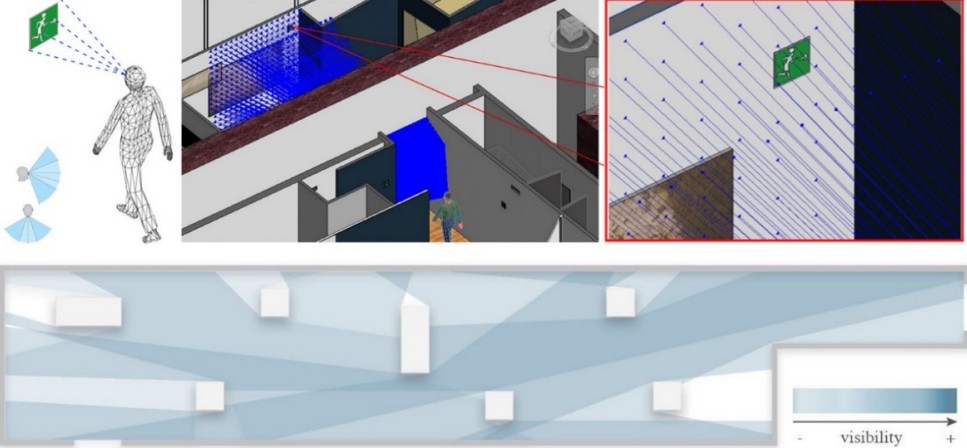

**Figure 7.** Verification of the safety signal visibility in the BIM environment. Orthographic and spatial views.

## 4. Criticalities and Adopted Workflows

The proposed approach considered the different disciplinary areas as interconnected elements, describing the "museum system" in its entirety, thanks to a geometrical and semantic characterization of the objects that make up the architectural and structural part, i.e., the plant systems (electrical, air conditioning, and security) and the collections.

Because the area of experimentation has a limited extension equal to about one thousand square meters, we opted to organize the different disciplines through the use of a single 3D model. In the first phases of the research, particular attention was paid to the historical and archival research for a more conscientious reconstructive digital modeling, which was followed by the BIM digitization of the CAD drawings produced during the recent intervention by Isolarchitetti.

The conversion between CAD files into BIM models involved intermediate steps to verify the accuracy and reliability of the information contained in the drawings, which may contain inconsistencies or do not report the as-built update. In particular, for the architectural part, inaccuracies were found that inevitably required a direct traditional survey. The limitation of errors was necessary both to virtually simulate the correct setting up of the exhibition "Invisible Archaeology" and to ensure the reliability of the information contained in the digital model (Figure 8).

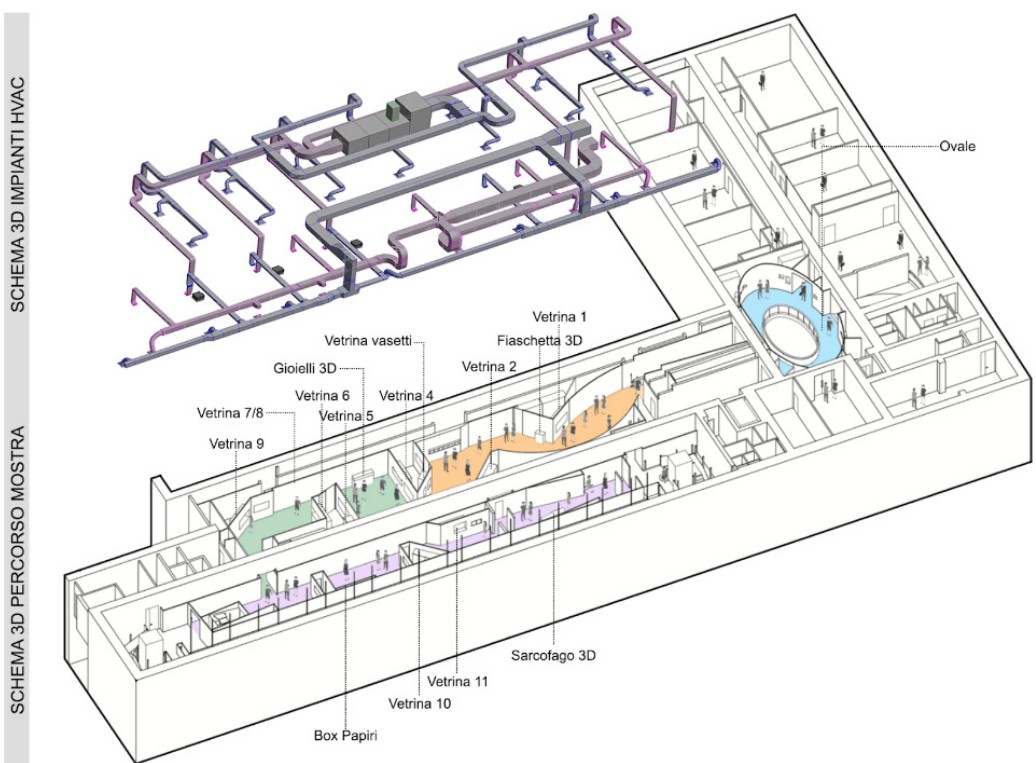

**Figure 8.** Overlapping of the architectural and plant models. "Invisible Archaeology" temporary exhibition. Exploded axonometric view made by Arch. M. Cammarano.

In relation to the plant components, some "hard clashes" interferences were recorded (Figure 9). These are characterized by the geometric overlapping of the air treatment plant with some of the tie bars of the metal carpentry. Similar to what happened on site, the overlaps were solved by considering the non-inspectionability of the air plant with the repositioning of some supply and return ducts.

The critical issue was the resolution of the wall-showcase joint. First, the wall was edited to create alcoves for the positioning of the exhibition screens. Second, considering the positioning of the display case between the uprights of the metal wall, the wall was

divided into parts in order to independently edit both the cladding, which has a different opening from that of the wall structure, and the structural layer.

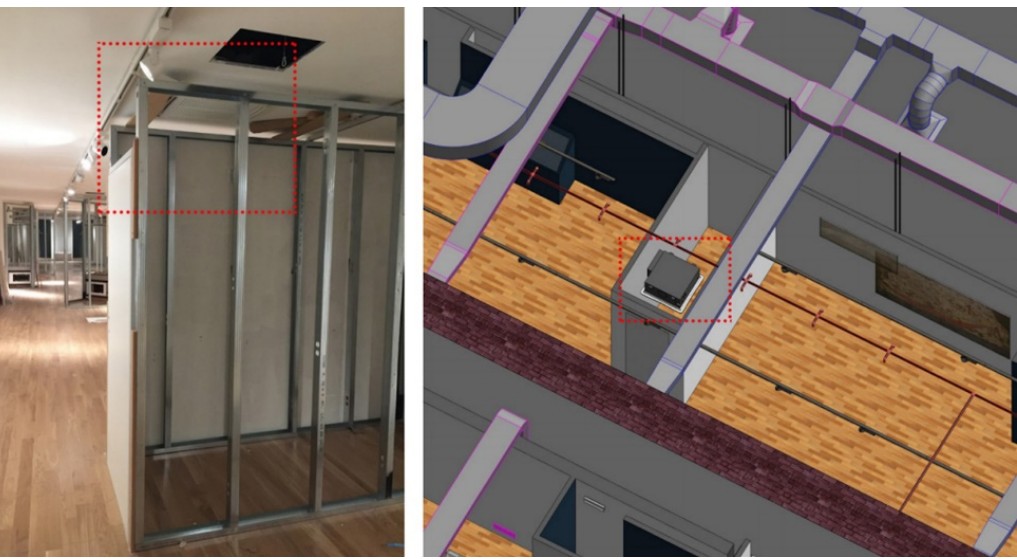

**Figure 9.** Clearance clash of the fan coil detected during the construction site phase of the "Invisible Archaeology" exhibition. Comparison between construction site and the BIM environment. Image by Arch. M. Cammarano.

In museums, the elements that host the collections, such as the showcases, play a fundamental role in guaranteeing adequate conditions for protective purposes and for efficient preventive conservation action through real-time control of the thermo-hygrometric values. What distinguishes the various showcases, except for the showcase for the vases of the funerary equipment of Kha and Merit, is the partitioning contained within them (Figure 10). The showcases have been recovered from the previous exhibition and reused by inserting new internal compartments.

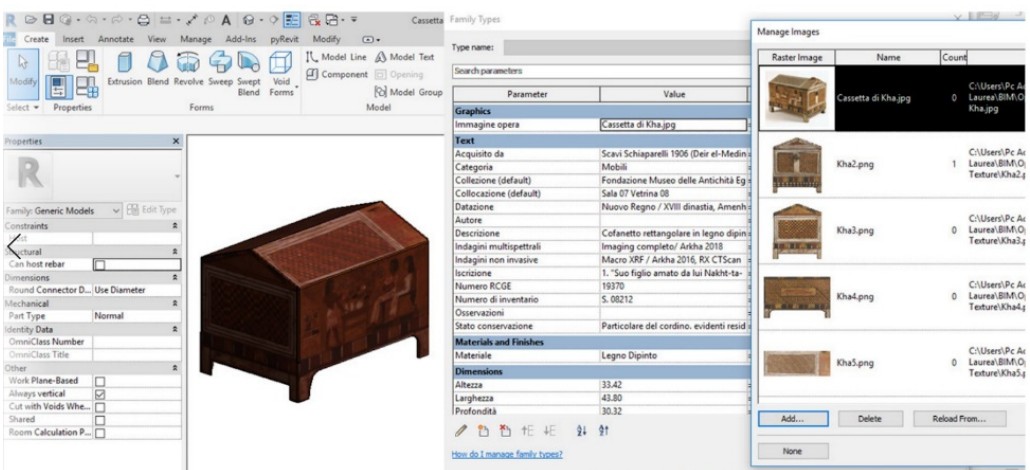

**Figure 10.** Collection models enriched by shared parameters. Cofanetto di Kha. Image by Arch. M. Cammarano.

Considering the hierarchical structure of the modeled components (families-types-instances) and the opportunity to reuse the showcases in the future, we proceeded with the creation of nested families of furniture systems.

## 5. Result

It is crucial to discuss the results obtained and how the algorithms were designed according to the "Begin with the end in mind" principle, setting up a workflow according to a specific system to achieve the desired result. To meet the needs of the Museum Management, the potential of BIM models was critically assessed. The adoption of graphic and alphanumeric databases capable of collecting information from various complex disciplines is an effective and efficient solution for management and monitoring interventions, representing an essential element for a periodic check on the maintenance status of the collections and building components that directly affect the conservation of the works of art. However, it immediately emerged that modeling the collection and museum system in a BIM environment presented some rigid intrinsic applications. In addition to the evident difficulties in modeling particular geometries, we considered the possibility of automating the compilation of the parameters associated with the various components, creating a bidirectional connection for the reading and writing of the database in editable spreadsheets. In particular, concerning the data related to the exhibits made available by the museum, there are more than two hundred thermo-parameters that must be directly related to the values of the exhibition spaces and the technical specifications of the showcases in charge of correct conservation and exhibition. To solve these criticalities, shared data management was proposed, and the disciplines that work together to manage and design spaces and museum collections were connected. To define this connection, it was decided to develop specific algorithms using a Visual Programming Language (VPL). Today, VPL is one of the tools used to enrich, with shared data, virtual models representing buildings and their collections, following the most well-known BIM procedures.

- Starting from the analysis suggested by the Ministerial Decree, the first research topic was identified, which led to the writing of a second algorithm relating to area VI-Collection Management. The script made it possible to evaluate the suitability of the collections for the host environment, taking into account the material composition of the objects. More generally, when assessing the feasibility of loans of artifacts between institutions, we referred to the Facility Report, a technical sheet containing the physical, environmental, and safety characteristics of a museum [19]. In this case, the information from the Facility Report has been attributed, and not based on the type of entity, and can be filtered and summarized, in a structured way, into specific data. Depending on the typological nature of the discipline to which it belongs, it is possible to interact with specific information relating to the single component by organizing or even dividing the dataset based on the permanent or temporary nature of the exposure. The platforms involved were Excel and Dynamo. The former was chosen especially for the majority of museum staff, who can follow the spreadsheets and update or implement the information associated with the entities of the digital model in a simplified and intuitive way. The use of the second was dictated by the experiment prototype, being a language that requires visual programming knowledge. It allowed for the construction of specific algorithms, even for personnel without specific IT skills, and it natively interfaced with the BIM applications used. The workflows identified within the Revit and VPL environment of Dynamo and Excel are still experimental, but they constitute a solid basis for setting up future applicative research on the use of these methodologies in the context of Cultural Heritage applied to museums.
- Regarding the development of the second tool for simulating user behavior within the museum space, the procedure, once automated, can support curators in controlling the design of the exhibition halls and possibly increase the quality of the objects on display. This procedure can be validated through the recording of user behavior for ex post analysis and the verification of congruence with the proposed preliminary phenomena. It is still necessary to recognize a methodology to correctly assign weights to single values, which can be mutual over time (Figure 11).

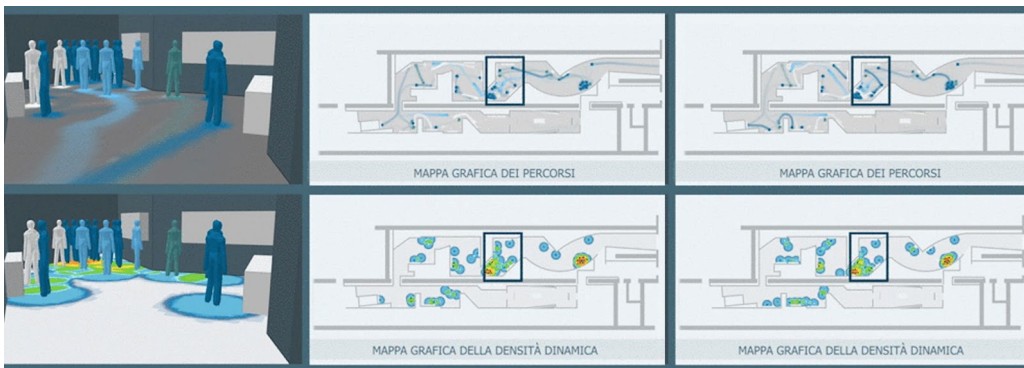

**Figure 11.** Simulation of different types of users in an exhibition space. On the first row, a graphic map of the paths; below, a graphic map of the dynamic density.

- A different algorithm was designed starting from the requirements of Legislative Decree 493/1996 [20], which allows one to evaluate the visibility of safety signs within an environment with reference to the furnishings present. Following studies carried out in the field of human vision, it has been found that the recognition of symbols and colors occurs within an angle of about 60° on the horizontal plane and about 55° on the sagittal plane. The geometric information was reproduced in the BIM environment using Dynamo tools. After setting the origin of the visual cone, a beam of vectors was generated to intersect the various elements of the virtual model, geometrically verifying that the safety signal was visible from the most significant number of selected points in the exhibition (Figure 12).

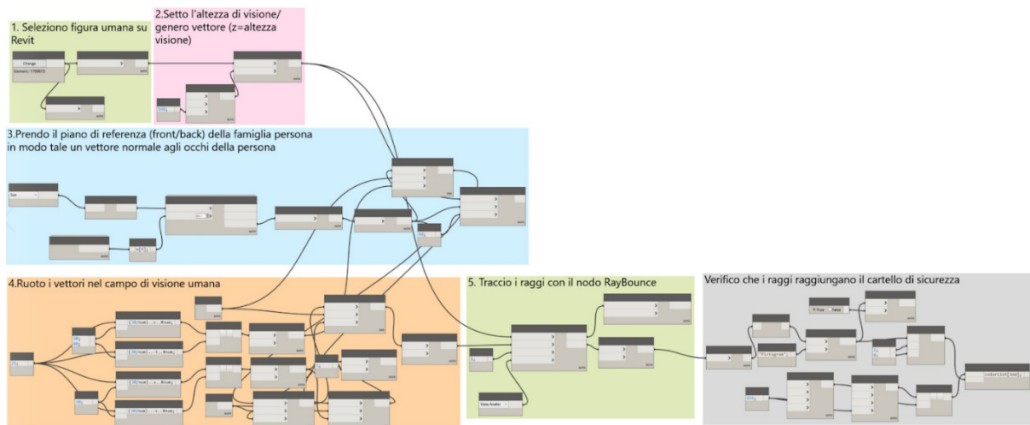

**Figure 12.** VPL algorithm for checking the visibility of safety signals. Image by Arch. M. Cammarano.

## 6. Conclusions

The research carried out showed how a BIM-oriented approach can be the bearer of innovation in the Museum Management domain. The collection of information coming from different disciplinary fields into a single model, not only regarding the AEC field but also concerning collections and exhibited artworks, can facilitate collaboration between the actors involved.

The research group focused on analyzing the theme of coordination between the various professional figures who collaborated in the management of a temporary exhibition. Gross errors and interferences can be identified and consequently limited through the adoption of a BIM methodology in the museum environment.

Furthermore, the functionalities of the digital model are constantly expanded due to the continuous development of VPL environments, which also allow not only architects and specialized personnel but "non-programmers" to write algorithms. In the museum field, for a correct safeguarding of heritage, the data to be found and analyzed should be

multiple and heterogeneous, affording an opportunity to expand the domain of application of VPL methodologies.

The research group focused on defining specific algorithms for the management and the accruement of these data. Still, although the results achieved in the visual programming environment can be considered positive, the adopted methodology could not be completely efficient for the generation of the entire computerized digital model of the building.

The methods presented are not intended to replace the experience of the professionals but to be a useful support for the work of curators and registrars. Once automated, these procedures can support the exhibitor to control the exhibition's design and eventually make it more efficient with respect to the topology and quality of the exhibits [21].

The methodologies applied during the research period still need to be refined and implemented; therefore, the future goals of the research group include the optimization of the process of construction of the museum digital model in a BIM environment and to the realization of tools that the museum staff can easily use, thus representing useful support to the delicate decision-making processes.

**Author Contributions:** Conceptualization, Massimiliano Lo Turco; methodology, Elisabetta Ca-terina Giovannini and Andrea Tomalini; software, Elisabetta Caterina Giovannini and Andrea Tomalini; validation, Elisabetta Caterina Giovannini and Andrea Tomalini; writing—original draft preparation, Massimiliano Lo Turco, Andrea Tomalini and Elisabetta Caterina Giovannini; writing—review & editing, Andrea Tomalini; project administration, Massimiliano Lo Turco.

**Funding:** This research received no external funding.

**Acknowledgments:** For the realization of the model in the BIM environment of the temporary exhibition, we thank Arch. Michele Cammarano; we also thank the "Associazione Amici Collaboratori del Museo Egizio" and the architect Andrea Megna, responsible for the Security of the Museo Egizio, for their great support and availability.

**Conflicts of Interest:** The authors declare no conflict of interest.

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
