# Peer review of "Parametric and Visual Programming BIM Applied to Museums, Linking Container and Content"

_ijgi, doi:10.3390/ijgi11070411_

Round 1

Reviewer 1 Report

The paper presents valuable research experiences and results that can significantly contribute to the sector. For this reason, I definitely recommend it for publication.

However, allow me to suggest a minor methodological revision. Throughout the paper, the authors deal with BIM and H-BIM terms (going even further - HS-BIM, etc.) and describe in detail applied workflows together with possibilities and benefits related to them, trying to propose a general approach. Nevertheless, it sounds to me that (as declared in the discussion of results) the paper reports on Revit-Dynamo (or anything Autodesk-related) applications. Nothing wrong with that: Autodesk products represent the choice of the majority of technicians working in the field. But terms such as families, global vs local models, etc. do apply only to the Autodesk ecosystem. And if we look at the topic from an open-BIM perspective (IFC) most differences (or even benefits) disappear. 

I would rather declare from the beginning how to collocate the paper. That would make the overall product stronger and more precise.

Author Response

Thanks for the review.

In the abstract we have explained the choice of tools and in the introduction of the proposed solutions we have motivated why we have chosen these tools. 

Reviewer 2 Report

This article deals with a remarkably interesting topic, namely the Parametric and visual programming BIM applied to museums (linking container and content collaborative learning experiences) in a changing environment regarding innovative approaches in Architecture.  

The article title “Parametric and visual programming BIM applied to museums, linking container and content” accurately reflects the content and purpose of the paper and it is absolutely integrated into "Heritage Building Information Modeling: Theory and Applications" Special issue of ISPRS International Journal of  Geo-Information.

This article reports and explores a synthesis of some experiences in which the research group has hybridized the Building Information Modelling methodologies with the Visual Programming Language systems. In this way they have been able to create tools for managing and controlling museum collections. The critical issues encountered in transposing the values of the methodologies proposed in the museum are also illustrated. This is without a doubt a work in progress presenting and discussing serious issues regarding well achieved experiences as well as other not so well achieved

The abstract is concise and provides sufficient information. The keywords are adequate. The introduction section presents a relatively good literature review and locates well the work. The article describes adequately the methodology and research methods. The results represent a contribution to a better knowledge of the new trends in monitoring the conditions in which a museum collection may be presented as well as, once automated, the devised procedures can be considered a prototype to support curators in controlling and improving the efficiency of the exhibition layout.

The bibliographic review and the references are important and up to date, but the consulted bibliography could still be improved and present a rather extensive reference development.

The figures should have a greater definition and print quality because it is an important part of the manuscript and actually it is very difficult to see or read some of the information contents.

The work is publishable, but it still needs these minor improvements before to be accepted. The article can be accepted if this is taken into consideration.

Author Response

Thanks for the review.

To better define the construction methodology of our models, we have cited some works previously conducted regarding the construction of families in HBIM environments.

To better define the methodology to manage IoT data, we cited a case study from which we initially drew inspiration, even if it does not interface with BIM systems.

Finally, we have explained why specific software has been chosen.

Reviewer 3 Report

1.     The study aims HBIM in a perspective concerning its application in a museum context, with an avatar path visualization and allowing visitors to consult of information about each element in exhibition. BIM and visual programing were linked contributing to enhance a virtual visit in a museum

2.     The background of study in well supported in several study cases pointing the problems of representing with accurate the geometry and the materials in order to create new families of HBIM parametric objects. In special the images of figure 2 represents, in  a very interesting diagram, the model process of specific window. In addition, the overlapping of HVAC system and the museum space illustrate clearly how to study, over a unique BIM model, the confits between two specialities.

3.     The study is very interesting and useful for the study case museum and can be replicated in other museums.

Author Response

Thanks for the review.

On the advice of other reviewers, we made small changes to explain why certain software was chosen and we integrated the bibliography to better introduce how we arrived at the definition of this methodology.
